# Misspecified Phase Retrieval with Generative Priors

**Zhaoqiang Liu**
National University of Singapore
dcslizha@nus.edu.sg

**Xinshao Wang**
University of Oxford
xinshao.wang@eng.ox.ac.uk

**Jiulong Liu**
Chinese Academy of Sciences
jiulongliu@lsec.cc.ac.cn

## Abstract

In this paper, we study phase retrieval under model misspecification and generative priors. In particular, we aim to estimate an $n$-dimensional signal $\mathbf{x}$ from $m$ i.i.d. realizations of the single index model $y = f(\mathbf{a}^T\mathbf{x})$, where $f$ is an unknown and possibly random nonlinear link function and $\mathbf{a} \in \mathbb{R}^n$ is a standard Gaussian vector. We make the assumption $\mathrm{Cov}[y, (\mathbf{a}^T\mathbf{x})^2] \neq 0$, which corresponds to the misspecified phase retrieval problem. In addition, the underlying signal $\mathbf{x}$ is assumed to lie in the range of an $L$-Lipschitz continuous generative model with bounded $k$-dimensional inputs. We propose a two-step approach, for which the first step plays the role of spectral initialization and the second step refines the estimated vector produced by the first step iteratively. We show that both steps enjoy a statistical rate of order $\sqrt{(k \log L) \cdot (\log m)/m}$ under suitable conditions. Experiments on image datasets are performed to demonstrate that our approach performs on par with or even significantly outperforms several competing methods.

## 1 Introduction

Compressed sensing (CS) is perhaps the most popular instance of high-dimensional inverse problems, for which one has the *linear* measurement model

$$y = \mathbf{a}^T\mathbf{x} + \eta, \tag{1}$$

where $\mathbf{a} \in \mathbb{R}^n$ is the sensing vector, $\mathbf{x} \in \mathbb{R}^n$ is the sparse signal to estimate, and $\eta$ represents additive noise. It has been well-known for CS that roughly $m = O(s \log(n/s))$ i.i.d. Gaussian measurements are sufficient to ensure the accurate recovery of a signal with $s$ non-zero entries [88, 1, 23, 11, 77].

Phase retrieval (PR) arises in numerous scientific areas including X-ray crystallography, acoustics, astronomy, microscopy, optics, wireless communications, and quantum information [13], where one cannot measure $\mathbf{a}^T\mathbf{x}$ directly, and can only record its magnitude. For example, the following *noisy magnitude-only* measurement model has been adopted in various prior works on sparse (real-valued) PR [92, 39, 25, 36, 9, 7]:

$$y = |\mathbf{a}^T\mathbf{x}| + \eta, \tag{2}$$

where the signal $\mathbf{x}$ is assumed to be sparse, and the sensing vector $\mathbf{a}$ is assumed to be a standard Gaussian vector.

However, both the linear and magnitude-only measure models in (1) and (2) are idealized views of the data generating process. To make the setup more general, one can utilize the following semi-parametric *single index model (SIM)* for general nonlinear models:

$$y = f(\mathbf{a}^T\mathbf{x}), \tag{3}$$

36th Conference on Neural Information Processing Systems (NeurIPS 2022).

where $f : \mathbb{R} \to \mathbb{R}$ is an *unknown* (possibly random) nonlinear link function, and $\mathbf{a}$ is typically assumed to be Gaussian. In addition, since the norm of $\mathbf{x} \in \mathbb{R}^n$ is absorbed into the SIM, for brevity, it is common to assume that $\mathbf{x}$ is a unit vector, i.e., $\|\mathbf{x}\|_2 = 1$. SIMs have long been studied in the conventional setting where the number of measurements $m > n$ [30, 81, 45]. In recent years, they have also been analyzed in the high-dimensional setting where $m \ll n$, mainly under the assumption that the underlying signal is sparse. Relevant works include but are not limited to [72, 63, 26, 73, 68]. For all these works, it is crucial to impose the following assumption on the SIM:

$$\mathrm{Cov}[y, \mathbf{a}^T \mathbf{x}] \neq 0. \tag{4}$$

The pivotal condition in (4) is fairly generic and encompasses notable special examples such as noisy 1-bit measurements and general (non-binary) quantization schemes. However, it is not satisfied by PR models including (2) and the related models

$$y = |\mathbf{a}^T \mathbf{x} + \eta|, \quad y = (\mathbf{a}^T \mathbf{x})^2 + \eta, \tag{5}$$

where $\eta$ refers to zero-mean random noise that is independent of $\mathbf{a}$. In order to formalize a class of SIMs that encompass the above-mentioned PR models (and more general models, see the discussion in Section 2.2) as special cases, misspecified phase retrieval (MPR) has been studied in [64, 98], with the condition (4) being replaced by the assumption

$$\mathrm{Cov}[y, (\mathbf{a}^T \mathbf{x})^2] \neq 0. \tag{6}$$

It is worth mentioning that another motivation behind studying MPR is that the theoretical analysis for PR typically relies on the correct model specification that the data points are indeed generated by the correct model, and the MPR model enables theoretical analysis under statistical model misspecification.

In both works [64, 98], the signal $\mathbf{x}$ is assumed to be sparse. Recently, motivated by tremendous successful applications of deep generative models and following the seminal work [6] on generative model based linear CS, it has been popular to study high-dimensional inverse problems under generative priors [76, 31, 32, 34, 96, 41, 2, 67, 95, 60, 40, 65]. In particular, instead of being sparse, the underlying signal is assumed to be contained in (or lie near) the range of a generative model. It has been empirically demonstrated in [6] and its various follow-up works (e.g., see [19, 85, 48] and a literature review in [78]) that compared to sparsity based methods, corresponding generative model based algorithms require significantly fewer samples to recover the signal up to a given accuracy.

In this paper, we study the MPR problem under generative modeling assumptions.

## 1.1 Related Work

In this subsection, we summarize some relevant works on PR and SIM, for both cases with or without generative priors.

**PR and SIM without generative priors**: There is a large amount of literature providing practical algorithms for PR, including convex methods [66, 14, 12, 90, 4, 29] and empirically more competitive non-convex methods [27, 21, 62, 70]. In particular, in the seminal work [62] and a variety of its follow-up works [13, 10, 15, 92, 91, 99, 39, 82, 8], whether it is for general PR (with no priors on the signal) or sparse PR, two-step approaches have been proposed with provable guarantees. The first step consists of a spectral initialization method, and the second step is typically an iterative (such as alternating minimization and gradient descent) algorithm that further refines the initial guess of the first step. For general PR, the spectral initialization step turns out to be unnecessary, and optimal sample complexity guarantees can be established even when using random initialization [83, 16]. However, to the best of our knowledge, it is not the case for sparse PR. More specifically, all theoretically-guaranteed non-convex algorithms for sparse PR require spectral initialization, and this typically results in a sub-optimal sample complexity of $O(s^2 \log n)$ (instead of $O(s \log n)$), where $s$ refers to the number of non-zero entries of the signal to estimate.

MPR is also closely related to SIMs, which have been extensively studied in the conventional setting where $m > n$, see, e.g., [30, 35, 59]. High-dimensional SIMs have received a lot of attention in recent years, with theoretical guarantees for signal estimation and support recovery [22, 24, 74, 84, 55, 72, 73, 17, 97, 28, 93, 69, 20]. In particular, motivated by the idea that under the assumption (4), a SIM can be converted into a scaled linear measurement model with unconventional noise, the authors

of [72] consider minimizing the linear least-squares objective function over a convex set. They show that a reliable estimation of the signal can be obtained by such a simple method despite the unknown nonlinear link function.

As mentioned earlier, MPR with sparse priors has been studied in [64, 98]. The work [64] implements a two-step procedure based on convex optimization, with the first step being a spectral initialization method. More specifically, a semidefinite program is solved to produce an initial vector, and then an $\ell_1$ regularized least square is solved to obtain a refined estimator. Such a procedure suffers from high computational costs. A more efficient two-step approach, which is a simple variant of the thresholded Wirtinger flow method [10], is proposed in [98]. In the first step, identical to that in [10], the initial vector is calculated by a thresholded spectral method that first estimates the support of the sparse signal by thresholding and then performs the classic power method over the submatrix corresponding to the estimated support. In the second step, a thresholded gradient descent algorithm is employed. Both approaches in [64, 98] can attain the optimal statistical rate of order $\sqrt{(s \log n)/m}$, provided that the sensing vector is standard Gaussian and the number of samples $m = \Omega(s^2 \log n)$. In addition, the second step of the approach proposed in [98] is shown to achieve a linear convergence rate.

**PR and SIM with generative priors**: PR with generative priors has been studied in [31, 37, 38, 80, 3, 47]. More specifically, an approximate message passing algorithm is proposed in [3]. The authors of [31, 80] minimize the objective over the latent space in $\mathbb{R}^k$ using gradient descent, where $k$ is the latent dimension of the generative prior. Corresponding algorithms may easily get stuck in local minima since the explorable solution space is limited. Recovery guarantees for projected gradient descent algorithms over the ambient space in $\mathbb{R}^n$ for noiseless PR with pre-trained or untrained neural network priors have been proposed in [37, 38]. No initialization methods have been proposed in these works, making the assumption on the initial vector therein a bit stringent. On the other hand, the authors of [47] propose a spectral initialization method for PR with generative priors and provide recovery guarantees with respect to globally optimal solutions of a corresponding optimization problem. The optimization problem is non-convex, and a projected power method is proposed in [50] to approximately find an optimal solution.

Generative model based SIMs have been studied in [51, 49, 46, 94]. The authors of [51, 49, 46] provide optimal sample complexity upper bounds under the assumption of Gaussian sensing vectors. But their results rely on the assumption (4), which is not satisfied by widely adopted PR models. The SIM studied in [94] encompasses certain PR models as special cases, and the sensing vector can be non-Gaussian. However, the nonlinear link function $f$ is assumed to be differentiable, making it not directly applicable to the PR model in (2). Moreover, it is worth mentioning that in both works [94, 51], the recovery guarantees are with respect to globally optimal solutions of typically highly non-convex optimization problems. Attaining these optimal solutions is practically difficult.

## 1.2 Contributions

Throughout this paper, we assume that the signal $\mathbf{x}$ lies in the range of an $L$-Lipschitz continuous generative model with bounded $k$-dimensional inputs. Our main contributions are as follows:

- We propose a two-step approach for MPR with generative priors. In particular, in the first step, we make use of the projected power method proposed in [50] to obtain a good initial vector for the iterative algorithm used in the second step.
- We show that under appropriate initialization, both steps attain a statistical rate of order $\sqrt{(k \log L) \cdot (\log m)/m}$, and the second step achieves a linear convergence rate. The initialization condition for the first step is mild in the sense that it allows the inner product between the starting point and the signal $\mathbf{x}$ to be a sufficiently small positive constant. In contrary, the initialization condition for the second step is more restrictive, making the first step necessary. Notably, unlike for the works on MPR with sparse priors [64, 98] that require the sub-optimal $O(s^2 \log n)$ sample complexity, the sample complexity requirement for our recovery guarantees is $O((k \log L) \cdot (\log m))$, which is naturally conjectured to be near-optimal [52, 42].
- We perform numerical experiments on image datasets to corroborate our theoretical results. In particular, for the noisy magnitude-only measurement model (2), we observe that our approach gives reconstructions that are competitive with those of the alternating phase projected gradient descent (APPGD) algorithm proposed in [37], which is the corresponding

state-of-the-art method, though we do not utilize the knowledge of the link function and allow for model misspecification. Moreover, for several closely related measurement models that satisfy the condition (6), our approach leads to superior reconstruction performance compared to all other competing methods, including APPGD.

## 2 Problem Formulation

In this section, we overview some important assumptions that we adopt. Before proceeding, we present the notation used in this paper.

### 2.1 Notation

We use upper and lower case boldface letters to denote matrices and vectors respectively. For any positive integer $N$, we use the shorthand notation $[N] = \{1, 2, \cdots, N\}$, and $\mathbf{I}_N$ represents the identity matrix in $\mathbb{R}^{N \times N}$. The support (set) of a vector is the index set of its non-zero entries. We use $\|\mathbf{X}\|_{2 \to 2}$ to represent the spectral norm of $\mathbf{X}$. We use $B^k(r)$ to denote the radius-$r$ in $\mathbb{R}^k$, i.e., $B^k(r) := \{\mathbf{z} \in \mathbb{R}^k : \|\mathbf{z}\|_2 \le r\}$, and $\mathcal{S}^{n-1}$ represents the unit sphere in $\mathbb{R}^n$, i.e., $\mathcal{S}^{n-1} := \{\mathbf{s} \in \mathbb{R}^n : \|\mathbf{s}\|_2 = 1\}$. We use $G$ to denote a pre-trained $L$-Lipschitz continuous generative model from $B^k(r)$ to $\mathbb{R}^n$. We focus on the setting where $k \ll n$. For any set $S \subseteq B^k(r)$, we write $G(S) = \{G(\mathbf{z}) : \mathbf{z} \in S\}$. Our goal is to estimate the signal $\mathbf{x} \in \text{Range}(G) = G(B^k(r))$ from realizations of the sensing vector $\mathbf{a}$ and the observation $y$ (generated according to the SIM in (3)). For any two sequences of real values $\{a_j\}$ and $\{b_j\}$, we write $a_j = O(b_j)$ if there exist an absolute constant $C_1$ and a positive integer $j_1$ such that for any $j > j_1$, $|a_j| \le C_1 b_j$, $a_j = \Omega(b_j)$ if there exist an absolute constant $C_2$ and a positive integer $j_2$ such that for any $j > j_2$, $|a_j| \ge C_2 b_j$. We use the generic notations $C$ and $C'$ to denote large positive constants, and we use $c$ to denote a small positive constant; their values may differ from line to line.

### 2.2 Setup

First, the following are the standard definitions of a sub-exponential random variable and the associated sub-exponential norm.

**Definition 1.** *A random variable $X$ is said to be sub-exponential if there exists a positive constant $C$ such that $(\mathbb{E}\left[|X|^p\right])^{1/p} \le Cp$ for all $p \ge 1$, or equivalently, if there exists a positive constant $C'$ such that $\mathbb{P}(|X| > u) \le \exp(1 - u/C')$ for all $u \ge 0$. The sub-exponential norm of $X$ is defined as $\|X\|_{\psi_1} := \sup_{p \ge 1} p^{-1} \left(\mathbb{E}\left[|X|^p\right]\right)^{1/p}$.*

We will focus on the following settings except where stated otherwise:

- The observations are independently generated according to the SIM (3), where $f$ is the link function that is *unknown* and possibly random.

- We have an $L$-Lipschitz continuous generative model $G : B^k(r) \to \mathbb{R}^n$. For convenience, similarly to [48, 50], we assume that the generative model is normalized such that $\text{Range}(G) \subseteq \mathcal{S}^{n-1}$. For a general (unnormalized) generative model, we may essentially consider its normalized version. See, e.g., [50, Remark 1].

- The signal $\mathbf{x}$ is contained in the range of $G$, i.e., $\mathbf{x} \in \text{Range}(G) \subseteq \mathcal{S}^{n-1}$.

- The sensing vector $\mathbf{a} \in \mathbb{R}^n$ is standard Gaussian, i.e., $\mathbf{a} \sim \mathcal{N}(\mathbf{0}, \mathbf{I}_n)$.

- The random variable $y = f(\mathbf{a}^T \mathbf{x})$ is sub-exponential with the sub-exponential norm being denoted by $K_y$, i.e., $K_y := \|y\|_{\psi_1}$. In addition, we use $M_y$ to denote the expectation of $y$, i.e., $M_y := \mathbb{E}[y]$.

  **Remark 1.** *$y$ will be sub-exponential when $f(x)$ comprises of $x^c$ plus lower order terms with $c \le 2$ (since the product of two sub-Gaussian random variables is sub-exponential), and therefore we will see that the $y$ corresponding to all the measurement models presented in our paper is sub-exponential. We remark that the assumption of sub-exponential $y$ is not essential and it can be easily relaxed. For example, when $y = x^c$ with $c$ being an even integer that is larger than 2, there will be only a minor change in the order of the $\log m$ term in the sample complexity and statistical rate. However, for brevity, we follow [64, 98] and make the assumption of sub-exponential $y$ to avoid non-essential complications.*

- We consider MPR and assume that[1]

$$\nu := \mathrm{Cov}\left[y, (\mathbf{a}^T \mathbf{x})^2\right] > 0, \tag{7}$$

or equivalently,

$$\nu := \mathrm{Cov}\left[f(g), g^2\right] > 0, \tag{8}$$

where $g \sim \mathcal{N}(0, 1)$ is a standard normal random variable. Note that the assumption (8) is only with respect to the nonlinear link function $f$. The condition in (7) is satisfied by PR models described in (2) and (5). It is also satisfied by relevant models such as [98]

$$y = |\mathbf{a}^T \mathbf{x}| + 2\tanh(|\mathbf{a}^T \mathbf{x}|) + \eta, \quad y = 2(\mathbf{a}^T \mathbf{x})^2 + 3\sin(|\mathbf{a}^T \mathbf{x}|) + \eta. \tag{9}$$

See [64, Proposition 3 and Remark 4] for more general examples.

# 3   Algorithm

In this section, we describe our two-step algorithm devised for MPR with generative priors. Suppose that we have $m$ i.i.d. realizations of $\mathbf{a}$ and $y$, namely $\mathbf{a}_1, \ldots, \mathbf{a}_m$ and $y_1, \ldots, y_m$. To estimate the signal $\mathbf{x}$, we consider the following two-step approach:

1. We perform $T_1$ iterations in the first step. In particular, let

$$\mathbf{V} = \frac{1}{m} \sum_{i=1}^{m} y_i \left(\mathbf{a}_i \mathbf{a}_i^T - \mathbf{I}_n\right). \tag{10}$$

We perform the projected power method proposed in [50]: For $t = 0, 1, \ldots, T_1 - 1$, let

$$\mathbf{w}^{(t+1)} = \mathcal{P}_G\left(\mathbf{V}\mathbf{w}^{(t)}\right), \tag{11}$$

where $\mathcal{P}_G(\cdot)$ denotes the projection function onto $\mathrm{Range}(G)$,[2] and we obtain $\mathbf{x}^{(0)} := \mathbf{w}^{(T_1)}$. Similarly to [43, 47], we set the starting point $\mathbf{w}^{(0)}$ as the column of $\frac{1}{m}\sum_{i=1}^{m} y_i \mathbf{a}_i \mathbf{a}_i^T$ (i.e., a shifted version of $\mathbf{V}$) that corresponds to the largest diagonal entry. Note that it is easy to calculate that $\mathbb{E}[\mathbf{V}] = \nu \mathbf{x}\mathbf{x}^T$ (see, e.g., [47, Lemma 8]), for which each column is a scalar product of $\mathbf{x}$. This motivates the use of a shifted version of $\mathbf{V}$ to get the initialization vector.

2. We perform $T_2$ iterations in the second step. In particular, let

$$\bar{y} = \frac{1}{m} \sum_{i=1}^{m} y_i \tag{12}$$

be the empirical mean of the observations. We perform the following iterative procedure: For $t = 0, 1, 2, \ldots, T_2 - 1$, let

$$\hat{\nu}^{(t)} = \frac{1}{m} \sum_{i=1}^{m} (y_i - \bar{y}) \cdot \left(\mathbf{a}_i^T \mathbf{x}^{(t)}\right)^2 \tag{13}$$

$$\tilde{y}_i^{(t)} = (y_i - \bar{y}) \cdot \left(\mathbf{a}_i^T \mathbf{x}^{(t)}\right), i = 1, 2, \ldots, m. \tag{14}$$

$$\tilde{\mathbf{x}}^{(t+1)} = \mathbf{x}^{(t)} - \frac{\zeta}{m} \cdot \sum_{i=1}^{m} \left(\hat{\nu}^{(t)} \cdot \left(\mathbf{a}_i^T \mathbf{x}^{(t)}\right) - \tilde{y}_i^{(t)}\right) \mathbf{a}_i, \tag{15}$$

$$\mathbf{x}^{(t+1)} = \mathcal{P}_G\left(\tilde{\mathbf{x}}^{(t+1)}\right), \tag{16}$$

where $\zeta > 0$ is a tuning parameter, and $\hat{\nu}^{(t)}$ can be thought of as an approximation of $\nu$ defined in (7). The idea behind calculating $\tilde{y}_i^{(t)}$ is that by comparing (4) and (6), we observe that to transform the MPR model into a conventional SIM, we may use $(y - \mathbb{E}[y])(\mathbf{a}^T \mathbf{x})$ to

---

[1]The case that $\nu < 0$ can be similarly handled by considering $-y$.

[2]That is, for any $\mathbf{s} \in \mathbb{R}^n$, $\mathcal{P}_G(\mathbf{s}) := G\left(\arg\min_{\mathbf{z} \in B^k(r)} \|G(\mathbf{z}) - \mathbf{s}\|_2\right)$. Similarly to [79, 37, 38, 71, 50], we implicitly assume the exact projection in our analysis. In practice, approximate methods such as gradient- and GAN-based projections [79, 75] have been shown to work well.

---

**Algorithm 1** A two-step approach for misspecified phase retrieval with generative priors

---

**Input**: $\{(\mathbf{a}_i, y_i)\}_{i=1}^m$, step size $\zeta > 0$, number of iterations $T_1$ for the first step, number of iterations $T_2$ for the second step, pre-trained generative model $G$, initial vector $\mathbf{w}^{(0)}$

**First step**:

  1: **for** $t = 0, 1, \ldots, T_1 - 1$ **do**
  2:     $\mathbf{w}^{(t+1)} = \mathcal{P}_G(\mathbf{V}\mathbf{w}^{(t)})$
  3: **end for**

**Second step**:
Let $\mathbf{x}^{(0)} := \mathbf{w}^{(T_1)}$

  1: **for** $t = 0, 1, \ldots, T_2 - 1$ **do**
  2:     Calculate $\hat{\nu}^{(t)}, \tilde{y}_i^{(t)}, \tilde{\mathbf{x}}^{(t+1)}$ and $\mathbf{x}^{(t+1)}$ according to (13), (14), (15), and (16), respectively
  3: **end for**

**Output**: $\hat{\mathbf{x}} := \mathbf{x}^{(T_2)}$

---

replace $y$, see also [64]. Moreover, (15) can be considered as a gradient descent step with respect to a linear measurement model with the scale factor $\hat{\nu}^{(t)}$ and observations $\tilde{y}_i^{(t)}$. Then, in (16), we project the calculated vector $\tilde{\mathbf{x}}^{(t+1)}$ onto $\text{Range}(G)$. Finally, we use $\hat{\mathbf{x}} := \mathbf{x}^{(T_2)}$ to represent the estimated vector obtained after $T_2$ iterations.

For convenience, we summarize the details in Algorithm 1.

## 4 Theoretical Results

The following theorem establishes recovery guarantees for the first step of Algorithm 1. The proof is given in the supplementary material. Note that $K_y := \|y\|_{\psi_1}$ (*cf.* Section 2.2) is considered a fixed constant and is omitted in the $O(\cdot)$ notation.

**Theorem 1.** *Assume that there exists a positive integer $t_0$ such that $\mathbf{x}^T \mathbf{w}^{(t_0)} \geq c_0$, where $c_0$ is a sufficiently small positive constant. Suppose that $m = \Omega((k \log(nLr)) \cdot (\log m))$ with a large enough implied constant. Then, we have that with probability $1 - O(1/m)$, it holds for all $t > t_0$ that*

$$\left\| \mathbf{w}^{(t)} - \mathbf{x} \right\|_2 \leq \frac{CK_y}{c_0} \sqrt{\frac{(k \log(nLr)) \cdot (\log m)}{m}} = O\left( \sqrt{\frac{(k \log(nLr)) \cdot (\log m)}{m}} \right). \quad (17)$$

Since a $d$-layer feedforward neural network generative model from $B^k(r)$ to $\mathbb{R}^n$ is typically $L$-Lipschitz continuous with $L = n^{\Theta(d)}$ [6] and we may set $r = n^{\Theta(d)}$, the upper bound in (17) is of order $\sqrt{(k \log L) \cdot (\log m)/m}$. Such a statistical rate is naturally conjectured to be near-optimal according to information-theoretic lower bounds established for MPR with sparse priors [64] and generative model based principal component analysis [50]. Therefore, Theorem 1 reveals that the first step of Algorithm 1 attains the near-optimal statistical rate under appropriate initialization and exact projections. The accurate projection assumption is perhaps the major caveat to Theorem 1. However, it is a standard assumption in relevant works including [79, 37, 38, 71, 50]. In practice, both gradient- and GAN-based projection methods [79, 75] have been shown to be highly effective in approximating the projection step.

**Remark 2.** *Spectral initialization steps in relevant works on* sparsity *based PR [92, 39, 25, 36, 9, 7] or MPR [64, 98] require the* sub-optimal *sample complexity $O(s^2 \log n)$, where $s$ refers to the number of non-zero entries. In contrary, according to Theorem 1, our spectral initialization step only requires the* near-optimal $O((k \log L) \cdot (\log m))$ *sample complexity (with a linear rather than a quadratic dependence on $k$). However, we note that such an advantage of our spectral initialization step comes at a price. In particular, we require the initialization condition $\mathbf{x}^T \mathbf{w}^{(t_0)} \geq c_0$, which is not required by spectral initialization steps in the above-mentioned works on sparse PR/MPR.*

**Remark 3.** *For some applications, we may assume that the dataset contains only vectors whose elements are all non-negative. For example, this is a natural assumption for image datasets. During pre-training, we can easily set the activation function of the last layer of the neural network generative model to be a certain non-negative function such as ReLU or sigmoid, and the range of such a*

*generative model is contained in the non-negative orthant. Therefore, the assumption that $\mathbf{x}^T \mathbf{w}^{(t_0)} \geq c_0$ for a sufficiently small positive constant $c_0$ is also mild. Similar assumptions have been made in relevant works including [18, 50] where it is not appropriate to assume that $-\mathbf{x}$ is also contained in the structured set (such as a closed convex cone or the range of a deep generative model). As a result, we provide an upper bound on $\|\mathbf{w}^{(t)} - \mathbf{x}\|_2$, instead of $\min\{\|\mathbf{w}^{(t)} - \mathbf{x}\|_2, \|\mathbf{w}^{(t)} + \mathbf{x}\|_2\}$, which is a commonly adopted distance measure in relevant literature on real-valued PR.*

Moreover, although the projected power iterations in the first step of Algorithm 1 can attain the near-optimal statistical rate under appropriate conditions, it is evident in a large body of literature on PR (see, e.g., [62, 13, 10, 99, 98, 64]) that such a spectral method better serves as the initialization step of a subsequent iterative approach. This motivates us to propose the second step of Algorithm 1, and in our numerical experiments, we clearly observe the benefit of the second step. More specifically, compared to simply performing the projected power method, performing both steps of Algorithm 1 leads to significantly better reconstructed images when the total number of iterations is fixed to be the same, namely $T_1 + T_2$.

Next, we present the following theorem, which is proved in the supplementary material. This theorem shows that under appropriate initialization and the assumption of exact projections, the iterative algorithm in the second step of Algorithm 1 converges linearly to a point achieving the near-optimal statistical rate of order $\sqrt{(k \log L) \cdot (\log m)/m}$.

**Theorem 2.** *Assume that the step size $\zeta$ and $\mathbf{x}^{(0)}$, which is the initial vector for the second step of Algorithm 1, satisfy*

$$2 \cdot |1 - \zeta\nu| + 5\zeta\nu \cdot \left\|\mathbf{x}^{(0)} - \mathbf{x}\right\|_2 + \beta_1 = 1 - \beta_2, \tag{18}$$

*where both $\beta_1$ and $\beta_2$ are positive constants.[3] Suppose that $m = \Omega((k \log(nLr)) \cdot (\log m))$ with a large enough implied constant. Then, we have with probability $1 - O(1/m)$, the following event occurs: There exists a positive integer $T_0 = O\left(\log\left(\frac{m}{(k \log(nLr)) \cdot (\log m)}\right)\right)$, such that the sequence $\left\{\|\mathbf{x}^{(t)} - \mathbf{x}\|_2\right\}_{t \in [0, T_0]}$ is monotonically decreasing, with the following inequality holds for all $t \leq T_0$:*

$$\left\|\mathbf{x}^{(t)} - \mathbf{x}\right\|_2 < (1 - \beta_2)^t \cdot \left\|\mathbf{x}^{(0)} - \mathbf{x}\right\|_2 + \frac{CK_y}{\beta_2}\sqrt{\frac{(k \log(nLr)) \cdot (\log m)}{m}}. \tag{19}$$

*In addition, we have for all $t \geq T_0$ that*

$$\left\|\mathbf{x}^{(t)} - \mathbf{x}\right\|_2 \leq \frac{CK_y}{\beta_2}\sqrt{\frac{(k \log(nLr)) \cdot (\log m)}{m}} = O\left(\sqrt{\frac{(k \log(nLr)) \cdot (\log m)}{m}}\right). \tag{20}$$

**Remark 4.** *In our analysis, we need to impose the assumption (18) on the step size $\zeta$ and initial vector $\mathbf{x}^{(0)}$. This makes the first step of Algorithm 1 necessary since when $\|\mathbf{x}^{(0)} - \mathbf{x}\|_2$ is not small, say $\|\mathbf{x}^{(0)} - \mathbf{x}\|_2 = 1$, the condition (18) cannot be satisfied. In comparison, to attain the near-optimal statistical rate $O(\sqrt{(k \log L) \cdot (\log m)/m})$, the initialization condition of the first step of Algorithm 1 is milder, and $\mathbf{x}^T \mathbf{w}^{(t_0)}$ (see Theorem 1) only needs to be lower bounded by a sufficiently small positive constant (thus $\|\mathbf{w}^{(t_0)} - \mathbf{x}\|_2$ can be close to $\sqrt{2}$). However, although the second step of Algorithm 1 requires a more restrictive initialization condition, we observe from our experimental results that it clearly refines the estimate of the first step. Such a phenomenon is also observed in various works related to PR, including [62, 13, 10, 99, 98, 64].*

**Remark 5.** *In Remark 4, we have briefly discussed the comparison of the initialization condition $\mathbf{x}^T \mathbf{w}^{(t_0)} \geq c_0$ in Theorem 1 and the typical initialization condition $\|\mathbf{x} - \mathbf{w}^{(t_0)}\|_2 < \delta\|\mathbf{x}\|_2$. In the following, we provide a more detailed discussion: When both $\mathbf{x}$ and $\mathbf{w}^{(t_0)}$ are unit vectors (this is the setting of our Theorem 1), the typical initialization requirement $\|\mathbf{x} - \mathbf{w}^{(t_0)}\|_2 < \delta\|\mathbf{x}\|_2$ can be reduced to $2\left(1 - \mathbf{x}^T \mathbf{w}^{(t_0)}\right) < \delta^2$, or equivalently, $\mathbf{x}^T \mathbf{w}^{(t_0)} 1 - \frac{\delta^2}{2}$. Note that $\delta$ is typically a small positive constant (e.g., $\delta = \frac{1}{6}$ in [10] and $\delta = \frac{1}{8}$ in [13]), and thus the typical initialization condition requires $\mathbf{x}^T \mathbf{w}^{(t_0)}$ to be larger than some positive constant that is close to 1. This is stronger than the assumption $\mathbf{x}^T \mathbf{w}^{(t_0)} \geq c_0$,[4] where $c_0$ is a sufficiently small positive constant.*

---

[3]$\beta_1$ is sufficiently small, and it is used to absorb a certain $o(1)$ term.

[4]It basically assumes the weak recovery of the signal, see, e.g., [61].

**Remark 6.** *The condition in* (18) *requires* $|1 - \zeta\nu| < \frac{1}{2}$. *This reveals that we should choose* $\zeta$ *such that* $\zeta \in \left(\frac{1}{2\nu}, \frac{3}{2\nu}\right)$, *and a good choice of* $\zeta$ *is* $\zeta = \frac{1}{\nu}$ *(for this case, the condition* (18) *reduces to* $\|\mathbf{x}^{(0)} - \mathbf{x}\|_2 < \frac{1}{5}$). *Since the knowledge of the link function* $f$ *(and thus the knowledge of* $\nu$, *which is dependent on* $f$; *See* (8)*) cannot be assumed, in our experiments, we use* $\hat{\nu}^{(t)}$ *to approximate* $\nu$. *That is,* $\zeta$ *is set to be* $\frac{1}{\hat{\nu}^{(t)}}$ *in the* $t$-*th iteration of the second step of Algorithm 1 (though it is slightly varying instead of being fixed).*

## 5 Numerical Results

In this section, we demonstrate the empirical performance of our Algorithm 1 (denoted by MPRG). We remark that these numerical experiments are proof-of-concept rather than seeking to be comprehensive since our contributions are primarily theoretical. We present some numerical results for the MNIST [44] dataset in the main document. Additional results for the MNIST dataset and some experimental results for the CelebA [53] dataset are presented in the supplementary material.

The MNIST dataset contains $60,000$ images of handwritten digits. The size of each image is $28 \times 28$, and thus the dimension of the image vector is $n = 784$. For the MNIST dataset, the generative model $G$ is set to be (the normalized version of) a pre-trained variational autoencoder (VAE) model with the latent dimension being $k = 20$. We make use of the VAE model pre-trained by the authors of [6] directly. For this VAE model, both the encoder and decoder are set to be fully connected neural networks with two hidden layers, and the architecture is $20 - 500 - 500 - 784$. The VAE model is trained by the Adam optimizer with a mini-batch size $100$ and a learning rate of $0.001$, and is trained from the images in the training set. The projection step $\mathcal{P}_G(\cdot)$ (*cf.* (11)) is approximated by the Adam optimizer with a learning rate of $0.1$ and $120$ steps.

We report the results on $10$ testing images that are selected from the test set. Note that these images are unseen by the pre-trained generative model. We perform $10$ random restarts. For reconstructed images, we choose the best among these $10$ random restarts to reduce the impact of local minima. We also provide quantitative comparisons with respect to the reconstruction error $\|\hat{\mathbf{x}} - \mathbf{x}\|_2$, where $\mathbf{x}$ is the underlying signal and $\hat{\mathbf{x}}$ refers to the estimated (normalized) vector produced by each of the methods described below. The reconstruction error is averaged over both the $10$ testing images and $10$ random restarts. All experiments are run using Python 3.6 and TensorFlow 1.5.0, with a NVIDIA GeForce GTX 1080 Ti 11GB GPU.

For Algorithm 1, we set $T_1 = 20$ and $T_2 = 30$. As mentioned in Section 3, the starting point $\mathbf{w}^{(0)}$ is set to be the column of $\frac{1}{m}\sum_{i=1}^{m} y_i \mathbf{a}_i \mathbf{a}_i^T$ (i.e., a shifted version of $\mathbf{V}$ defined in (10)) that corresponds to the largest diagonal entry. In addition, as mentioned in Remark 6, we set the step size $\zeta$ as $\zeta = \frac{1}{\hat{\nu}^{(t)}}$ (*cf.* (13)) in the $t$-th iteration of the second step of Algorithm 1. We compare our Algorithm 1 (denoted by MPRG) with the following methods: 1) The method proposed in [98], which is for misspecified phase retrieval with sparse priors and is denoted by MPRS. All the involved parameters are set to be the same as those in [98]. 2) Simply performing the first step of Algorithm 1 for $T_1 + T_2 = 50$ iterations. The corresponding method is denoted by PPower. The purpose of comparing to PPower is to verify the benefit of the second step of Algorithm 1. 3) Simply performing the second step of Algorithm 1 for $T_1 + T_2 = 50$ iterations. The corresponding method is denoted by Step2. The purpose of comparing to Step2 is to check whether the first step of Algorithm 1 is practically useful. 4) The Alternating Phase Projected Gradient Descent (denoted by APPGD) algorithm proposed in [37]. This algorithm is specifically designed for phase retrieval with magnitude-only measurements (*cf.* (2)) and generative priors, and the corresponding iterative procedure is

$$\mathbf{x}^{(t+1)} = \mathcal{P}_G\left(\mathbf{x}^{(t)} - \frac{\tau}{m}\sum_{i=1}^{m}\left(\left(\mathbf{a}_i^T\mathbf{x}^{(t)}\right) - y_i \cdot \mathrm{sign}\left(\mathbf{a}_i^T\mathbf{x}^{(t)}\right)\right)\mathbf{a}_i\right), \tag{21}$$

where $\tau > 0$ is the step size. We follow [37] to set $\tau = 0.9$. For a fair comparison, we use the vector produced after $T_1 = 20$ iterations of the first step of Algorithm 1 as the initialization vector of APPGD, and then we run APPGD for $T_2 = 30$ iterations.

We first consider the noisy magnitude-only measurement models for $i \in [m]$,

$$y_i = |\mathbf{a}_i^T\mathbf{x}| + \eta_i, \tag{22}$$

$$y_i = |\mathbf{a}_i^T\mathbf{x} + \eta_i|, \tag{23}$$

where $\eta_i$ are i.i.d. realizations of an $\mathcal{N}(0, \sigma^2)$ random variable. For such a measurement model, the corresponding random nonlinear link function $f$ is $f(x) = |x| + \eta$ or $f(x) = |x + \eta|$, where $\eta \sim \mathcal{N}(0, \sigma^2)$. The numerical results for (22) and (23) are demonstrated in Figures 1 and 2. From these figures, we observe that the sparsity based method MPRS attains poor reconstructions, and the results of PPower are not desirable. The three methods Step2, APPGD, MPRG lead to similar reconstruction error, but the reconstructed images of APPGD and MPRG are better than those of Step2. In particular, our method MPRG leads to mostly accurate reconstructions that are competitive compared to those of APPGD, even if we do not make use of the knowledge of the link function $f$ and MPRG is not specifically designed for the magnitude-only measurement models.

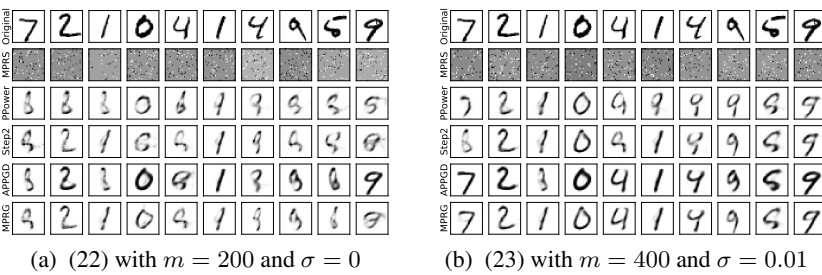

(a) (22) with $m = 200$ and $\sigma = 0$        (b) (23) with $m = 400$ and $\sigma = 0.01$

Figure 1: Examples of reconstructed MNIST images for the measurement models (22) and (23).

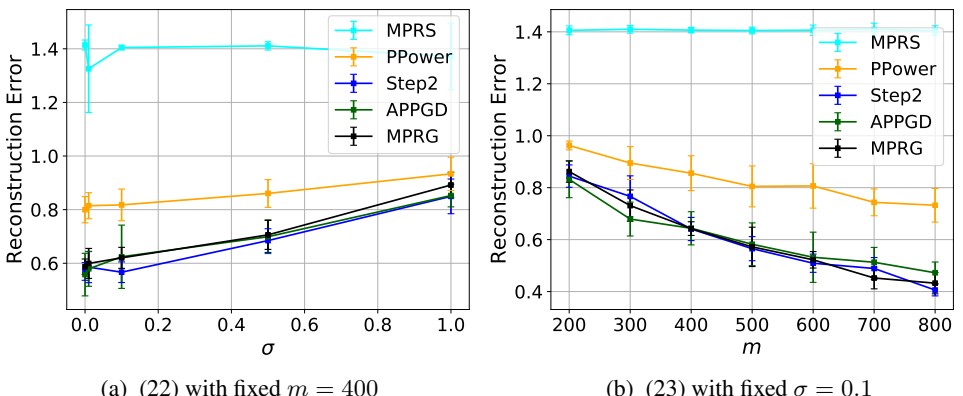

(a) (22) with fixed $m = 400$        (b) (23) with fixed $\sigma = 0.1$

Figure 2: Quantitative comparisons of the performance for the measurement model (22) and (23).

Next, we consider the following two measurement models:

$$y_i = |\mathbf{a}_i^T \mathbf{x}| + 2\tanh(|\mathbf{a}_i^T \mathbf{x}|) + \eta_i, \tag{24}$$

$$y_i = 2(\mathbf{a}_i^T \mathbf{x})^2 + 3\sin(|\mathbf{a}_i^T \mathbf{x}|) + \eta_i, \tag{25}$$

where again $\eta_i$ are i.i.d. realizations of an $\mathcal{N}(0, \sigma^2)$ random variable. For both models in (24) and (25), the corresponding link functions satisfy the condition (8) for MPR [98]. The numerical results are presented in Figures 3 and 4. We observe from these figures that for the measurement models (24) and (25), our method MPRG achieves the best reconstructions. In particular, it outperforms APPGD in terms of recovery quality and/or reconstruction error.

## 6 Conclusion and Future Work

We have proposed a two-step approach for phase retrieval under model misspecification and generative priors. We show that under suitable conditions, both steps of our approach obtain estimated vectors that achieve the near-optimal statistical rate of order $\sqrt{(k \log L) \cdot (\log m)/m}$, where $k$ is the latent dimension and $L$ is the Lipschitz constant of the generative model respectively, and $m$ refers to the number of samples.

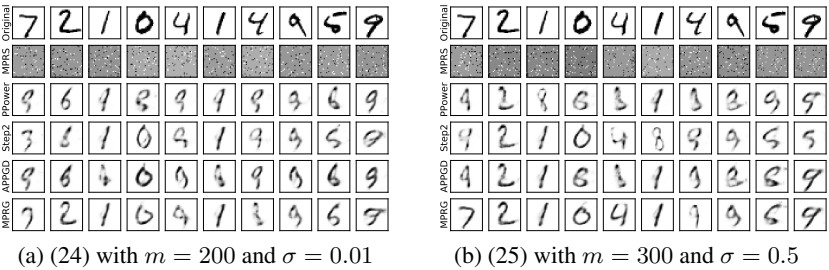

(a) (24) with $m = 200$ and $\sigma = 0.01$      (b) (25) with $m = 300$ and $\sigma = 0.5$

Figure 3: Examples of reconstructed MNIST images for the models in (24) and (25).

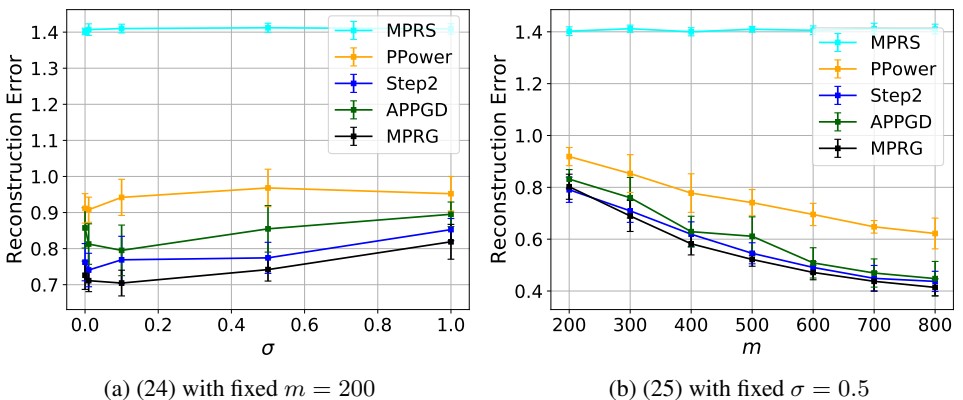

(a) (24) with fixed $m = 200$      (b) (25) with fixed $\sigma = 0.5$

Figure 4: Quantitative comparisons of the performance for the measurement models in (24) and (25).

We assume accurate projections in our analysis and use a gradient-based method to approximate the projection step $\mathcal{P}_G(\cdot)$ in our experiments. Although the exact projection assumption is commonly made in relevant works [79, 37, 38, 71, 50], it is a very interesting future research direction to design provably-guaranteed efficient methods for the projection step.

In addition, we focus on real Gaussian measurements. While we believe that based on the technical results in [62, 13] (which study complex Gaussian measurements), it is straightforward to extend our work to the complex case, the extension to more practical non-Gaussian measurement models such as sub-sampled Fourier measurements is a very interesting future direction. Another direction is to use different preprocessing functions to enhance the performance of our spectral initialization method [56, 54, 57]. Moreover, if one has the access to the nonlinear link function $f$, the Bayes-optimal performances can characterized using message-passing algorithms [5, 58, 3]. It would be interesting to connect or compare our results with the corresponding Bayes-optimal rate.

**Acknowledgment.** J. Liu was partially supported by NSFC Grant #12288201 and the Fund of the Youth Innovation Promotion Association, CAS (2022002). We sincerely thank the five anonymous reviewers and area chair for their careful reading and helpful comments. We are extremely grateful to Dr. Jonathan Scarlett for proofreading the manuscript and giving valuable suggestions.

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
