# OpenReview forum: "Misspecified Phase Retrieval with Generative Priors"
_NeurIPS.cc/2022/Conference — NeurIPS 2022 Accept_

### Official Review · Reviewer_c4oX · 2022-07-01

**Rating:** 6
**Confidence:** 3
**Soundness:** 4 excellent
**Presentation:** 3 good
**Contribution:** 2 fair

**Summary:**

This work considers the phase retrieval problem. In this model, the statistician is given
$m$ i.i.d. observations of $y = f(\mathbf{a}^T \mathbf{x})$, in which $\mathbf{a} \sim \mathcal{N}(0,\mathrm{I}_n)$, and $\mathbf{x} \in \mathbb{R}^n$ (the vector to recover) is assumed to be generated using a generative
prior $G : \mathbb{R}^k \to \mathbb{R}^n$ (typically with $k \ll n$).
Crucially, she does not know the function $f$, a setting which the authors call misspecified phase retrieval.
Under the following fairly general condition on the function $f$: $\mathrm{Cov}[f(g), g^2] \neq 0$,
for $g \sim \mathcal{N}(0,1)$, the authors develop in Section 3 an algorithm that is agnostic to the function $f$, and that is made of two steps:
- First, a general spectral method on a well-chosen matrix, mixed with a projection on the range of the generative model.
- Secondly, a simple iterative algorithm approximating procedures used in other single-index models, again completed with a projection step.

In Section 4, the authors provide two theorems giving guarantees on the performance of each step of the algorithm described above,
and argue that under certain conditions they reach near-optimal performance.

Finally, the authors provide numerical experiments in Section 5 to illustrate how their procedure performs with respect to some previous algorithms.

Please note that given the length and available time, I did not check in full detail the proofs given in the supplementary material, and only read them superficially.
I also read the Section D of the supplementary on additional numerical experiments. However, I did not look at the provided code.

**Questions:**

1. From my understanding, the condition $\mathrm{Cov}[f(g), g^2] \neq 0$ is crucial for the spectral method, since it ensures that for $m$ large enough, an isolated eigenvalue
(with eigenvector in the direction of $\mathbf{x}$) pops out in the spectrum of $\mathbf{V}$. Is this the only point in which this condition is necessary?
This point should be discussed in relation with another question I have on the spectral method used, see below.

2. The paper does not discuss a recent literature on spectral methods for phase retrieval,
cf. e.g. [3-6] for studies in the case of Gaussian vectors $\mathbf{a}_i$.
In particular, [4,6] shows that, in the case in which $f(g)$ is known and $\mathbf{x}$ does not come from a generative prior,
the optimal spectral method is as in $(10)$, in which $y_i$ is replaced by a function $T^\star(y_i)$, with e.g.\ $T^\star(y) = 1 - y^{-1}$ for noiseless magnitude observations. Moreover,
in this case, recovery can be achieved for $m = \Theta(n)$, rather than $m = \Omega(n \log^2 n)$ given by Theorem 1 here.
While this setting has significant differences from the one considered here (in particular $T^\star(y)$ is not accessible), maybe the authors should also consider the impact of the use of different functions $T(y) \neq y$ in the spectral method.
This might allow interesting comparisons with the literature mentioned, potentially improve drastically the performance of the method, as well as maybe relaxing the main condition assumed on $f(g)$ (see my point 1)?

[3] Mondelli, M., \& Montanari, A. (2018, July). Fundamental limits of weak recovery with applications to phase retrieval. In Conference On Learning Theory (pp. 1445-1450). PMLR.

[4] Luo, W., Alghamdi, W., \& Lu, Y. M. (2019). Optimal spectral initialization for signal recovery with applications to phase retrieval. IEEE Transactions on Signal Processing, 67(9), 2347-2356.

[5] Lu, Y. M., \& Li, G. (2020). Phase transitions of spectral initialization for high-dimensional non-convex estimation. Information and Inference: A Journal of the IMA, 9(3), 507-541.

[6] Maillard, A., Krzakala, F., Lu, Y. M., \& Zdeborová, L. (2022, April). Construction of optimal spectral methods in phase retrieval. In Mathematical and Scientific Machine Learning (pp. 693-720). PMLR.

3. If the statistician has access to $f$, the Bayes-optimal performances has been characterized using message-passing algorithms
in detail in [7-9] (in particular [9] tackles the case of generative models, and is mentioned in the paper).
Do the authors know how this Bayes-optimal error compares to the ones obtained in Figures 2 and 4?
It would be interesting to compute it, to gauge the influence of two effects: ($i$) the gain offered by the knowledge of $f$, and $(ii)$ the gain offered by the use of the generative model (e.g. by comparing the Bayes-optimal performance without a generative prior to the one reached by MPRG -- which is agnostic to $f$ and uses generative priors). In general, I believe the paper would benefit from discussing more the practical influence of the structure induced by the generative prior on the reconstruction error.

[7] Barbier, J., Krzakala, F., Macris, N., Miolane, L., \& Zdeborová, L. (2019). Optimal errors and phase transitions in high-dimensional generalized linear models. Proceedings of the National Academy of Sciences, 116(12), 5451-5460.

[8] Maillard, A., Loureiro, B., Krzakala, F., \& Zdeborová, L. (2020). Phase retrieval in high dimensions: Statistical and computational phase transitions. Advances in Neural Information Processing Systems, 33, 11071-11082.

[9] Aubin, B., Loureiro, B., Baker, A., Krzakala, F., \& Zdeborová, L. (2020, August). Exact asymptotics for phase retrieval and compressed sensing with random generative priors. In Mathematical and Scientific Machine Learning (pp. 55-73). PMLR.

4. In Remark 2: while I see that one might assume all coordinates of $\mathbf{x}$ to be positive,
could the authors clarify why they can assume that $\mathbf{w}^t$ has only positive coordinates?

5. This paper only considers real phase retrieval. Can the authors comment on the possibility of extension of their work to the (more relevant for many applications) complex case? It is likely that the results extends quite straightforwardly, which would be interesting to mention.

6. Finally, I found the following typos:
- Line 58: ``seminar work''
- Line 73: ``an SIM''
- Line 79: ``an $l_1$ regularized least square''
- Line 219: ``all non-negative vectors''?
- In Theorem 2, there is no mention of $\beta_1$ in both eqs. (19) and (20), some $\beta_2$ must surely be changed to a $\beta_1$ (at the end of (19) and in (20) ?)
- Line 698 (supplementary): ``there exists such a net with the cardinality satisfies''

**Limitations:**

Nothing to signal on societal impacts. On limitations of this work and how the authors address them, see the previous points.

**Strengths And Weaknesses:**

- I found the paper overall very well organized, as well as well-written and pleasant to read, and I thank the authors for that.
In general, the results are quite clearly stated and discussed.
Moreover, the limitations of the results are not hidden and well discussed, e.g. the projection step in the algorithm that needs to be approximated,
the Lipschitz condition on the generative model, the ``informative initialization'' assumption required in Theorem 1, or the additional $\log m$ in the rates obtained with respect to the optimal ones.

- The numerical results are well-presented and convincing. While the improvements over APPGD or the pure spectral method are small,
they seem to be significative.

- On the other hand, the initialization condition of Theorem 1 seems quite limiting: it basically assumes weak recovery of the signal, while usually spectral methods are used to obtain such a weakly-correlated estimator. The theorem only covers how the method improves from weak to strong correlations. While this does not seem to be an issue in practice (if I understood correctly, the authors do not force such a weak correlation to exist in the numerical simulations), it limits the scope of Theorem 1.

- Another criticism I have is that, from a non-specialist viewpoint, the results seem somehow incremental with respect to [1]: while I was not familiar with this paper before, a rapid read suggests that perhaps the main addition of the present paper is to consider generative models, which in the end only leads to minor changes in the algorithm with respect to Algorithm 1 of [1].  Moreover, the analysis of the spectral method relies on results of [2], and is quite classical in my eyes. Perhaps the authors should discuss more the novelties of this work with respect to such previous analyses (e.g. by an exploration of the gains offered in practice by generative priors, see also my question below).

[1] Neykov, M., Wang, Z., \& Liu, H. (2020). Agnostic estimation for misspecified phase retrieval models. The Journal of Machine Learning Research, 21(1), 4769-4807.

[2] Liu, Z., Liu, J., Ghosh, S., Han, J., \& Scarlett, J. (2022). Generative principal component analysis. arXiv preprint arXiv:2203.09693.

---

> ### Author Response · Authors · 2022-08-02
> **Responses to Reviewer c4oX**
>
> Thanks for your recognition of this paper and the insightful comments and suggestions. Our responses to the main concerns are as follows (the responses to your concerns about the initialization condition of Theorem 1, the novelties with respect to previous analyses, and the comparison to the Bayes-optimal error in [3] are provided in the general responses to all reviewers). All citations refer to the reference list in the submitted main document.
>
> (**About the condition $\mathrm{Cov}[f(g),g^2] \ne 0$**) We thank the reviewer for this insightful comment. We believe that the point mentioned by the reviewer corresponds to the fact that $\mathbb{E}[\mathbf{V}]= \nu \mathbf{x} \mathbf{x}^T$ (see the proof of Lemma 5), and we believe that this is the only point in which the condition $\nu := \mathrm{Cov}[f(g),g^2] \ne 0$ is necessary.
>
> (**The paper does not discuss recent literature on spectral methods for phase retrieval & The Bayes-optimal performances has been characterized using message-passing algorithms**) We thank the reviewer for pointing out these interesting papers to us and for the nice summary of the ideas of two of these papers. We agree that our paper would benefit from considering the impact of the use of different functions $T(y) \ne y$ in the spectral method and from discussing more the practical influence of the structure induced by the generative prior on the reconstruction error. However, we believe that addressing the reviewer's comments is orthogonal to our main goal, which is to provide recovery guarantees for MPR under generative priors. While the reviewer’s comments have value in refining and extending the general scope of work in the broader area (which would be a major research accomplishment in itself), that appears to be better left to a dedicated piece of work, and we will cite all the papers mentioned by the reviewer in the Conclusion and Future Work section in our revised version.
>
> (**Could the authors clarify why they can assume that $\mathbf{w}^{(t)}$ has only positive coordinates?**) The assumption that $\mathbf{w}^{(t)}$ has only non-negative coordinates is mild since $\mathbf{w}^{(t)} \in \mathrm{Range}(G)$ and we can easily set the activation function of the last layer of the pre-trained neural network generative model $G$ to be a certain non-negative function such as ReLU or sigmoid during pre-training (e.g., for the pre-trained VAE model used for the MNIST dataset, the last layer has sigmoid activation).
>
> (**Can the authors comment on the possibility of extension of their work to the (more relevant for many applications) complex case?**) Thanks for the suggestion. We believe that based on the technical results in [54, 12] (which study complex Gaussian measurements), it is straightforward to extend our work to the complex case, and we will mention this in the revised version.
>
> (**Typos**) We thank the reviewer for the careful reading of our paper and we will correct all the typos in the revised version. In particular, we will modify the sentence “We may assume that the dataset contains all non-negative vectors” to “We may assume that the dataset contains only vectors whose elements are all non-negative”. In addition, $\beta_1$ is a sufficiently small positive constant that is used to absorb a certain $o(1)$ term (see Eqs. (155) and (156)). To avoid confusing the reader, we will mention that it is a sufficiently small positive constant in the statement of Theorem 2.

---

> > ### Comment · Reviewer_c4oX · 2022-08-05
> > **Following on the discussion**
> >
> > I thank the authors for detailed answers. In general I have found them globally satisfactory, and I look forward to the changes to be implemented in the paper. On two of the points discussed:
> >
> > - I agree with the author's response that the work [3] tackles the Bayes-optimal performance in a more restricted class of models than the one studied in this paper, in a more specific regime. Still I think it would have been a good experiment to compare the performances of the algorithm presented here exactly in the setting of [3] (which, as you mentioned, is included in the setting considered here), to gauge -- even in a restricted setting -- how much one gains by having access to $f$. I agree nevertheless that this might require more work than what is expected in a review phase.
> >
> > - Coming back to the condition $\mathrm{Cov}[f(g),g^2] \neq 0$, since the only point in which it is necessary is to show that the spectral method matrix points in expectation in the direction of $\mathbf{x}$, one could also mention that using different preprocessing functions $\mathcal{T}(y)$, the condition becomes  $\mathrm{Cov}[\mathcal{T}[f(g)],g^2] \neq 0$. Thus it seems to me that this is no longer a restriction on the model considered, but only a mild condition on the preprocessing function chosen in the algorithmic procedure (which, on top of that, can greatly enhance the performance of the method, see my question 2). Perhaps this should also be mentioned in an opening paragraph on different preprocessing functions.

---

> > > ### Author Response · Authors · 2022-08-05
> > > **Follow-up responses to Reviewer c4oX**
> > >
> > > We are pleased that the reviewer found our answers globally satisfactory, and we thank the reviewer again for the comments. Our responses to the two points are as follows:
> > >
> > > (**Comparison with the Bayes-optimal performance**) This is a helpful suggestion. We will compare the performances of our algorithm and the AMP algorithm proposed in [3] exactly in the setting of [3].
> > >
> > > (**Different preprocessing functions**) This is also a helpful suggestion. We agree with the reviewer that by using different preprocessing functions $\mathcal{T}(y)$, the performance of our spectral initialization method can be greatly enhanced. In an opening paragraph on preprocessing functions, we will mention that when using different $\mathcal{T}(y)$, the condition becomes $\mathrm{Cov}[\mathcal{T}[f(g)],g^2] \ne 0$ (and the matrix $\mathbf{V}$ becomes $\frac{1}{m}\sum_{i=1}^m \mathcal{T}(y_i) (\mathbf{a}_i \mathbf{a}_i^T -\mathbf{I}_n)$).

---

### Official Review · Reviewer_YJ4f · 2022-07-03

**Rating:** 5
**Confidence:** 3
**Soundness:** 3 good
**Presentation:** 2 fair
**Contribution:** 3 good

**Summary:**

This work proposes a two-step algorithm for solving the misspecified phase retrieval problem. The algorithm is constructed based on two key assumptions:  a) the $m$ (possibly noisy) observations $y_{i}$ are generated by a single-index model $y_{i} = f(a_{i}^{\top}x)$ (a.k.a. generalised linear model) with $a\sim\mathcal{N}(0,I_{n})$, but *crucially* the statistician does not has access to the link function $f$ -  she only knows that the observations $y$ correlate with $(a^{\top}x)^2$; b) the signal $x$ is drawn from a *known* generative prior, i.e. $x=G(z)$ where $G:B(r)\to\mathcal{S}^{n-1}$ is $L$-Lipschitz and $z\in B(r)\subset\mathbb{R}^{k}$ is a latent representation ($B(r)$ is the ball radius $r$ and $\mathcal{S}^{n-1}$ the unit sphere). The algorithm consists of a spectral initialisation-type step, based on a projected power method, plus a descent-like step.

The main theoretical contributions are:

1. A bound on the reconstruction performance of the first step, stating that for sufficiently enough data $m = \Omega(k\log(nLr))$, the first step achieves near-optimal reconstruction performance $O(\sqrt{k\log(nLr)/m})$ (up to a log factor in $m$) with probability $1-O(1/m)$ (Theorem 1).
2. A bound on the reconstruction performance showing that for a suitable learning rate and warm initialisation (the reason why step 1 is required), the second step achieves near-optimal reconstruction with probability $1-O(1/m)$ for sufficient data $m = \Omega(k\log(nLr))$ (Theorem 2)

Numerical simulations with real datasets, different link functions and with trained generative models are used to illustrate the performance of the proposed algorithm, and to compare it with other methods in the literature.

**Questions:**

-**[Q1]**: The information about the generative model only appears in the bounds eqs. (17) and (18) through the Lipschitz constant $L$ and the radius $r$ (in particular their product $Lr$). Can the authors provide some intuition on how these quantities are connected with the expressiveness of the prior? For instance, for a fixed generative model architecture, how $L$ changes from random initialisation throughout the training? Can this result be used to help us choosing a specific prior for a given task?

-**[Q2]**: The bounds in Step 1 eq. (17) and Step 2 eq. (20) scale in the same way with the quantities $(L, r, m, n, k)$ involved in the problem. Is it clear that performing Step 2 always improve over Step 1 (i.e. $\beta_{2}>\beta_{1}$)? If not, did the authors observe any example in which Step 2 after Step 1 hurts reconstruction?

-**[Q3]**: In which sense is the rate $\sqrt{(k\log{L})(\log{m})/m}$ near-optimal? For a given generative prior, say a fully connected random network, how does this compares with the Bayes-optimal rate derived in [3]?

-**[Q4]**: How would the algorithm perform in the matched case, where the statistician would have access to the exact link function $f$ generating the observations $y$ (i.e. we could let $\hat{\nu}^{t}\to \nu$)? Can a tighter bound be derived for this case? Would it consistently beat APPGD?

-**[Q5]**: It would be interesting to have also an experiment with the link $y=|a^{\top}x + \eta|$ in Sec. 5, which is relevant to some experimental settings where $y\geq 0$.

**Limitations:**

Some limitations of this work are discussed, e.g. the fact that the theoretical guarantees assume exact knowledge of the projection $\mathcal{P}_{G}$ which in practice needs to be approximated and the choice of step size $\zeta$ for Step 2, which in practice also needs to be estimated by $1/\hat{\nu}^{t}$.

**Strengths And Weaknesses:**

Phase retrieval is an important problem naturally arising in different signal processing tasks from science and engineering [SEC+]. Although it is not very different from linear reconstruction problems from the point of view of information theory (both require $m\approx n$ samples for statistically reconstructing a Gaussian signal [MLK+]), it is a notably harder problem computationally. Therefore, designing algorithms that exploit structure in the signal for efficient reconstruction is a significant endeavour.

While many works have approached this problem by deriving algorithms for sparse signals (c.f. [SEC+] for a review), a more recent line of work have investigated the setting where the signal is drawn from a generative model, typically parametrised by a deep neural network (e.g. VAEs, GANs, etc.) [30], with promising computational advantages. This paper builds on this line, with the main difference with respect to the literature being the "misspecified" setting, i.e. the algorithm proposed does not rely on the knowledge of a particular link function for the observation likelihood.

**Strengths**:
The contribution is timely and well-placed within the literature. As discussed above, phase retrieval is a challenging computational problem relevant to different fields. Therefore the design of efficient algorithms that exploit structure with theoretical guarantees is a significant contribution to this line of work.

**Weaknesses**:
This work heavily builds on previous contributions - the algorithm proposed is a combination of methods from the literature (projected power method [45, 47] and projected descent [70]). While I don't think this is a problem per se, this makes the presentation hard to parse for a reader who is not familiar with this literature. For instance, the authors could discuss better where the proposed algorithm comes from, and provide some intuition for Steps 1 & 2. The authors could also be more explicit about what information is assumed to be known to the statistician in the derivation of the algorithm (the term "misspecified" is not very precise, and can mean different things in statistics).

**References** (numbered refs. are from the bibliography in the paper)

[SEC+] Y Shechtman, YC Eldar, O Cohen, HN Chapman, J Miao, M Segev, *Phase Retrieval with Application to Optical Imaging: A contemporary overview*, in IEEE Signal Processing Magazine, vol. 32, no. 3, pp. 87-109, May 2015.

[MLK+] A Maillard, B Loureiro, F Krzakala, L Zdeborová, *Phase retrieval in high dimensions: Statistical and computational phase transitions*, NeurIPS 2020.

---

> ### Author Response · Authors · 2022-08-02
> **Responses to Reviewer YJ4f**
>
> Thanks for your helpful comments and suggestions. Our responses to the main concerns are as follows (the responses about the technical novelties and the comparison with the Bayes-optimal rate derived in [3] are provided in the general responses to all reviewers).
>
> (**Intuition for Steps 1 & 2**) We thank the reviewer for the helpful suggestions. The major intuition for Step 1 is that even if we do not assume the knowledge of the link function $f$, we still have that the expectation of $\mathbf{V}$ is $\nu \mathbf{x}\mathbf{x}^T$ (see the proof of Lemma 5), for which $\mathbf{x}$ is the leading eigenvector. Since the classic power method is popular for obtaining the leading eigenvector and the underlying signal $\mathbf{x}$ is assumed to lie in the range of a generative model, we follow [47] to use a variant of the classic power method that projects the calculated vector onto the range of the generative model in each iteration.
>
> The major intuition for step 2 is that when using $(y − \mathbb{E}[y])(\mathbf{a}^T \mathbf{x})$ to replace $y$, the MPR model can be transformed into a conventional single index model that satisfies Eq. (4), and it can be further converted into a scaled linear measurement model with unconventional noise.
>
> We will add these intuitions into our revised version, and we will be more explicit about what information is assumed to be known to the statistician in the derivation of the algorithm.
>
> (**Q1**)  A $d$-layer feedforward neural network generative model is typically $L$-Lipschitz continuous with $L = n^{\Theta(d)}$ (see [5]) and we may set $r = n^{\Theta(d)}$. Studying how $L$ changes from random initialisation throughout the training would be quite interesting, but it is beyond the scope of this work.
>
>  (**Q2**) According to our experimental results, performing Step 2 always improves over Step 1, and we did not observe any example in which Step 2 after Step 1 hurts reconstruction.
>
> (**Q3**)  According to the algorithm-independent lower bound established for MPR with sparse priors (which is $\Omega(\sqrt{(s\log n)/m}$), where $s$ is the number of non-zero entries of the signal; see [56, Thm. 8]) and the algorithm-independent lower bound established for generative model based principal component analysis (which is $\Omega(\sqrt{(k\log L)/m})$; see Theorem 3 in the arXiv version of [47]), the rate $\sqrt{(k\log L)\cdot (\log m)/m}$ is naturally conjectured to be near-optimal (the $\log m$ term only plays a minor role).
>
> (**Q4**) If we have access to the exact link function $f$ (and thus the knowledge of $\nu$, which is only dependent on $f$), we believe that using the correct $\nu$ (instead of using $\hat{\nu}^{(t)}$) will lead to (at least slightly) better reconstruction performance (see the general responses about technical novelties compared to [70], where we mentioned that the scale factor plays an important role). We believe that the $\log m$ term can be removed (and thus lead to a tighter bound) for the case that we have access to the exact link function $f$.
>
> (**Q5**) Thanks for the comment. We will add the experimental results for $y = |\mathbf{a}^T \mathbf{x} +\eta|$ into Sec. 5 in our revised version.

---

### Official Review · Reviewer_NiZz · 2022-07-09

**Rating:** 6
**Confidence:** 3
**Soundness:** 3 good
**Presentation:** 2 fair
**Contribution:** 3 good

**Summary:**

The paper considers the problem of recovering a signal $x \in \mathbb{R}^n$ from $m$ measurements corresponding to a single index model of the form $y_i = f(a_i^\top x)$. Here, $a_i\in \mathbb{R}^n$ are sensing vectors and $f:\mathbb{R} \rightarrow \mathbb{R}$ is an unknown (may be random) non-linear function. The single index model encompasses the classical phase retrieval problem $y_i = |a_i^\top x| +\eta$ (among others), where $\eta$ is noise. Under the assumption on $f$ that $(a^\top x)^2$ and $y$ have non-null covariance, $x$ is in the range of a $L$-Lipschitz generative model $G:\mathbb{R}^k \rightarrow \mathbb{R}^n$ and suitable randomness on $a_i$ and $y_i$, the paper studies a two-stage algorithm for recovering $x$ given $(y_i )_{i=1}^m$ and $G$: the first step provides an initial estimate of $x$ using spectral methods and the second step iteratively refines this estimate by solving a related linear recovery problem using a descent-type method and projecting onto the range of the generator at each iteration.

**Questions:**

- Algorithm 1 part 1 uses the matrix V to get the initialization for part 2. Could you provide a intuition on what V is (or what the action of V on an arbitrary vector gives)?

- Figure 1a shows denoising property of your algorithm. Does the decay in reconstruction error match the theorem of $C/\sqrt{m}$ (for a fixed k)?

- The paper focuses on the case where the exact form of $f$ is not known. This is true for the numerical experiments presented as well. How would recovery of the MNIST images change if $f(x) = |x|$ and this information was used? (for example using the algorithms in [71] or [30]). Comparison of to these methods would be interesting.




**Limitations:**

The paper adequately addresses its limitations.


**Strengths And Weaknesses:**

Strengths:
- The paper studies an interesting problem.
- The paper builds on existing work on projected power method to obtain an estimate and introduces a new scheme to refine the estimate.
- The paper provides a statistically optimal rate of recovery on the order of $\sqrt{k/m}$ with near-optimal sample complexity of $O(k\log(L)\log(m))$.

Weakness:

- The paper is not clearly written. For example:
	- In the abstract, the notation Cov[.,.] should explained before using it.
	- In line 22, the "For example" line provides example of works that study methods of solving the phase retrieval problem while referencing (to the previous sentence) to applications of phase retrieval instead.
	- In line 31, the authors state that norm of $x$ is absorbed in SIM without an explanation (maybe stating that the problem is only well-defined up to that norm scaling would be beneficial to the readers).
	- The paper starts by associating Compressed Sensing unambiguously with the case where the underlying signal is sparse. However, this is misleading as the prior on signal can be arbitrary but low dimensional and the signal recovery problem can still be referred to as compressed sensing.

- The paper considers the case where the measurement matrix is random gaussian. A discussion on viability (or lack-thereof) of the proposed algorithm on a more realistic measurement models like sub-sampled Fourier would be useful for the readers.

- The paper provides limited discussion on the projection onto the range of the generative model under the conditions required for Theorem 2. That is, under the conditions like $L$-Lipschitz generative model, is there a guarantee that gradient based method as used in the experiment in the paper will converge to the global minimum/minima?

---

> ### Author Response · Authors · 2022-08-02
> **Responses to Reviewer NiZz**
>
> Thanks for your recognition of this paper and the useful comments and suggestions. Our responses to the main concerns are as follows (the response to your concern about Gaussian measurements is provided in the general responses to all reviewers).
>
> (**The paper is not clearly written**) We thank the reviewer for pointing out the problems in our writing. We will correct these problems in our revised paper.
>
> (**Is there a guarantee that gradient based method as used in the experiment in the paper will converge to the global minimum/minima?**) We conjecture that to guarantee the convergence to global minimum/minima for gradient based methods, we need to follow the analytic framework of [30] and assume a ReLU neural network with i.i.d. zero-mean Gaussian weights and no offsets. Although beyond our scope, this is a very interesting future direction.
>
> (**Could you provide intuition on what $\mathbf{V}$ is?**) The expectation of $\mathbf{V}$ is $\nu \mathbf{x}\mathbf{x}^T$ (see the proof of Lemma 5), for which each column is a scalar product of $\mathbf{x}$. This motivates the use of $\mathbf{V}$ (which is regarded as an approximation of $\nu \mathbf{x}\mathbf{x}^T$) to get the initialization vector. We will add such an intuition into the revised version.
>
> (**Does the decay in reconstruction error match the theorem of $C/\sqrt{m}$ (for a fixed $k$)?**) Thanks for the question. We will add the corresponding figures (with the $x$-axis being $1/\sqrt{m}$ and the $y$-axis being the reconstruction error) into the revised paper.
>
> (**How would recovery of the MNIST images change if $f(x) = |x|$ and this information was used? (for example using the algorithms in [71] or [30])**) It seems that due to additional assumptions adopted to deduce powerful theoretical guarantees on favorable optimization landscape, it is not suitable to compare with the algorithm proposed in [30]. (In particular, the authors of [30] make the assumption about a ReLU neural network generative model with i.i.d. zero-mean Gaussian weights and no offsets, which is not satisfied by our pre-trained neural network models. For example, for the pre-trained VAE model used for the MNIST dataset, the weights are not i.i.d. Gaussian, the activation function of the last layer is sigmoid (not ReLU), and there are offsets.) We will compare with the algorithm proposed in [71] for the case $f(x) = |x|$ (or the noisy version).

---

### Official Review · Reviewer_VWWv · 2022-07-10

**Rating:** 6
**Confidence:** 3
**Soundness:** 3 good
**Presentation:** 3 good
**Contribution:** 3 good

**Summary:**

The goal is to recover signals under the misspecified phase retrieval model, where measurements are generated $y_i = f(a_i^T x)$, $i = 1 \ldots m$, for i.i.d. Gaussian $a_i \in \mathbb{R}^n$, for $x$ generated by an $L$-lipschitz generative model, and for an unknown link function $f$. If $f$ satisfies the criteria $\text{Cov}_{g = a^Tx}[f(g), g^2] \not = 0$, then the proposed algorithm can be used to recover $x$ at order $O((k \log L) \cdot (\log m))$ measurements, conjectured to be near-optimal. This assumption is natural for the MPR problem as it captures any nonzero correlation between the misspecified measurements and the phaseless measurements in the correctly-specified phase retrieval model.

The proposed algorithm has two steps. The first step estimates the principal component of

 $V = \frac{1}{m} \sum_{i=1}^m y_i (a_i a_i^T - I_n)$, for which $\mathbb{E}_{a\sim \mathcal{N}}[V] = \nu x x^T$. The second step views misspecified phase retrieval as a conventional SIM with measurements $\tilde{y}_i = (y - \mathbb{E}[y])(a^T x)$, applying a projected gradient based method to iteratively improve the estimate from the first step. Both of these steps can be applied independently to solve MPR separately, with the first having weaker initialization assumptions. The authors find empirically that the second step improves estimates provided by the first step. Additionally, they prove that the first and second steps enjoy near-optimal statistical rates, and that the second step converges exponentially fast to an estimate below the guaranteed statistical error threshold.

**Questions:**

- In experiments using approximate projections onto the range of pretrained GANs, how quickly does step 2 converge? Is it that Theorem 2 holds approximately in experiments using approximate projections?
- What is the recovery performance when step 2 is run with initialization $\mathbf{w}^{(0)}$, the column with the largest diagonal entry in $\frac{1}{m} \sum_{i=1}^m y_i a_i a_i^T$? In other words, what is the performance of the combination step 1+2 relative to the performance of step 2 only? Is step 1 required to satisfy the initialization requirements for step 2, in practice? This question is related to my concern about the strictness/weakness of Theorem 2 assumptions.

**Limitations:**

- The method requires exact projection onto the range of a generator function. However, this seems to be a standard assumption in the literature, for which approximate methods work well.
- It is unclear whether the recovery condition in Theorem 2 is strict or mild. However, the benefit of step 2 is clearly beneficial empirically, whether or not Theorem 2 can be satisfied.

**Strengths And Weaknesses:**

Strengths:

- Theorems 1 and 2 are clearly stated and relevant to the empirical problem. I did not find any obvious errors while reviewing their proofs.
- Theorem 1 seems to be a small, but original and valuable contribution relative to [45] and [47]. The authors establish in Lemma 6 a bound on the error $E = V - \nu x x^T$, thereby controlling the misspecification error for the MPR problem. Following from the results of [47], Lemma 6 implies the estimation rate for step 1, giving theoretical justification to the practical method introduced in [45].
- To the best of my knowledge, exponential converge attained by Theorem 2 is original and valuable. Some results in the S-REC framework [5, 48] show that global optimizers of a certain non-convex objective achieve low estimation error, _without_ proving that this non-convex objective can be efficiently optimized. Therefore it is interesting that the gradient-based approach of step 2 has exponential convergence above the statistical error level.
- Empirical evaluations clearly demonstrate the significant benefit of the second step of the algorithm.
- Empirical evaluations are favorable to MPRG (proposed), which performs better than or on-par with alternative methods.

Weaknesses:

- It’s difficult to know the practical significance of the initialization requirements for Theorems 1 and 2, particularly the latter. Regarding theorem 1, Remark 2 is a convincing argument that the Theorem 1 requirement is mild. However, it’s unclear whether (18) is strict or mild, and whether there is any reason to believe it holds approximately in practice.

---

> ### Author Response · Authors · 2022-08-02
> **Responses to Reviewer VWWv**
>
> Thanks for your recognition of this paper and the useful comments and suggestions. Our responses to the main concerns are as follows.
>
> (**It's unclear whether (18) is strict or mild**) When $\zeta = 1/\nu$ (in the experiments, we need to use $\hat{\nu}^{(t)}$ to approximate $\nu$), (18) reduces to $\\|\mathbf{x}^{(0)} -\mathbf{x}\\|_2 < \frac{1}{5}$ (see Remark 4). This coincides with the condition $\mathrm{dist}(\mathbf{x}^{(0)},\mathbf{x}) < \delta \\|\mathbf{x}\\|_2$ (note that in our settings, both $\mathbf{x}$ and $\mathbf{x}^{(0)}$ are unit vectors and the distance measure is $\\|\mathbf{x}^{(0)} -\mathbf{x}\\|_2$), which is commonly used in relevant works (see, e.g., [12, Eq. (3.1)], [54, Thm. 4.1], and [89, Lem. 3.1]). Such a condition will be satisfied if the spectral initialization step returns an $\mathbf{x}^{(0)}$ that is close to $\mathbf{x}$.
>
> (**How quickly does step 2 converge? Is it that Theorem 2 holds approximately in experiments using approximate projections?**) In our experiments, we found that $T_2 = 30$ steps are usually sufficient for step 2 to converge. In our revised version, we will add the figures with the number of iterations of step 2 (namely $T_2$) being the $x$-axis and the reconstruction error being the $y$-axis. From our experimental results, we observe that step 2 works well and we believe that Theorem 2 holds approximately in experiments using approximate projections.
>
> (**Is step 1 required to satisfy the initialization requirements for step 2, in practice?**) This is a very interesting question. Step 1 is at least required to satisfy the initialization requirements for step 2 in theory (see Remark 3), and it has been standard to use a spectral initialization step in follow-up works of [54] to provide recovery guarantees for phase retrieval. We will perform the suggested experiments to check whether step 1 is also required to satisfy the initialization requirements for step 2 in practice and add the corresponding numerical results into our revised version.

---

### Official Review · Reviewer_nWLU · 2022-07-11

**Rating:** 5
**Confidence:** 3
**Soundness:** 4 excellent
**Presentation:** 3 good
**Contribution:** 3 good

**Summary:**

The authors study a single index model (SIM) for observations of the form $y = f(ax)$, with $f$ being unknown nonlinear function, $a$ is Gaussian, and $Cov[y,(ax)^2] \neq 0$. This is also referred as Misspecified phase retrieval (MPR). The inverse problem is solved under the assumption that $x$ has a generative prior. The overall algorithm is a two step approach which utilizes a spectral type initialization, followed by projected iterative descent rule using this initialization. The algorithm is validated via numerical simulations.

**Questions:**

Can the condition $x^T w^{t_0} > c_0$ be experimentally validated? What does $c_0$ typically look like under finite sampling.

What are some classes of functions $f$ that would satisfy $y = f(ax)$ being subexponential? The paper would benefit from additional discussion on the link function $f$.

Would the theoretical arguments fall through for non-Gaussian measurements?

**Limitations:**

Authors have made useful references to papers with potential overlap, as well as provided discussions on the limitations of their assumptions which is good.

**Strengths And Weaknesses:**

Strengths:

-Use of generative priors in the context of misspecified phase retrieval (MPR) under new set of assumptions.

-Theoretical assumptions and remarks are well presented.

Weaknesses:

-Not enough motivating discussion on why one needs to study Misspecified phase retrieval (largely left to reference papers).

-Gaussian measurements are not practical.

-Strong theoretical assumptions. Requires $x^T w^{t_0} > c_0$.

-How is $x^T w^{t_0} > c_0$ a weaker requirement than closeness of initialization requirement $\||x-w^{t_0}\|| < \delta \||x\||$ (for which spectral initialization is typically used)?

-Sample complexity comparison is unfair if $x^T w^{t_0} > c_0$ is used. Given proper initialization, even prior methods incur O(k) samples for generative priors or O(s) for sparse priors.

---

> ### Author Response · Authors · 2022-08-02
> **Responses to Reviewer nWLU**
>
> Thanks for your useful comments. Our responses to the main concerns are given as follows (the responses to your concerns about Gaussian measurements and strong theoretical assumptions are provided in the above general responses to all reviewers).
>
> (**Not enough motivating discussion on why one needs to study MPR**) The motivations mainly follow those discussed in reference papers [56,89]. In particular, the two major motivations are as follows:
>
> (i) The MPR model encompasses the noisy phase retrieval model as a special case in addition to various other additive and non-additive models with even link functions (see [56, Page 2]).
>
> (ii) Theoretical analysis for PR typically relies on the correct model specification that the data points are indeed generated by the correct model, and the MPR model enables theoretical analysis under statistical model misspecification (see [89, Page 2]).
>
> We will add these motivations into our revised paper, instead of simply leaving them to reference papers.
>
> (**Comparison of the two initialization conditions**) We have briefly discussed the comparison of $\mathbf{x}^T\mathbf{w}^{(t_{0})} \ge c_0$ and $\\|\\mathbf{x}-\mathbf{w}^{(t_0)}\\|_2 < \delta \\|\mathbf{x}\\|_2$ in Remark 3. In the following, we provide a more detailed discussion: When both $\mathbf{x}$ and $\mathbf{w}^{(t_0)}$ are unit vectors (this is the setting of our Theorem 1), the typical initialization requirement $\\|\\mathbf{x}-\mathbf{w}^{(t_0)}\\|_2 < \delta \\|\mathbf{x}\\|_2$ can be reduced to $2(1-  \mathbf{x}^T\mathbf{w}^{(t_0)}) < \delta^2$, or equivalently,  $\mathbf{x}^T\mathbf{w}^{(t_0)} > 1- \frac{\delta^2}{2}$. Note that $\delta$ is typically a small positive constant (e.g., $\delta = \frac{1}{6}$ in [9] and $\delta = \frac{1}{8}$ in [12]), and thus the typical initialization condition requires $\mathbf{x}^T\mathbf{w}^{(t_0)}$ to be larger than some positive constant that is close to $1$. This is stronger than the assumption in our Theorem 1, which requires $\mathbf{x}^T\mathbf{w}^{(t_0)} \ge c_0$ with $c_0$ being a sufficiently small positive constant. We will add such a discussion into our revised version.
>
> (**Sample complexity comparison is unfair if $\mathbf{x}^T\mathbf{w}^{(t_{0})} \ge c_0$ is used**) We agree that the sample complexity comparison is unfair if $\mathbf{x}^T\mathbf{w}^{(t_{0})} \ge c_0$ is used, and we have mentioned this in Remark 1 that "we note that such an advantage of our spectral initialization step comes at a price".
>
> (**Additional discussion on the link function $f$**) $y$ will be sub-exponential when $f(x) = x^c + \text{lower order terms}$ with $c \le 2$ (since the product of two sub-Gaussian random variables is sub-exponential), and therefore the $y$ corresponding to all the measurement models presented in our paper is sub-exponential. We remark that the assumption of sub-exponential $y$ is not essential and it can be easily relaxed (in fact, this assumption is mainly used in Eqs. (30), (32), and (41) in the supplementary material). For example, when $f(x) = x^c$ with $c$ being a positive and even integer that is larger than $2$, there will be only a minor change in the order of the $\log m$ term in the sample complexity and statistical rate. But for brevity, we follow [56,89] to make the assumption of sub-exponential $y$ to avoid non-essential complications. We will add these additional discussions on the link function $f$ into the revised version.

---

### Author Response · Authors · 2022-08-02
**General responses to the five anonymous reviewers**

We are very grateful to the reviewers for their helpful feedback and suggestions, and are pleased to have received a generally positive response. Our responses to the main concerns shared by multiple reviewers are given as follows. Other responses are given to each reviewer separately. All citations refer to the reference list in the submitted main document.

(**Gaussian measurements**) The assumption about Gaussian measurements is standard for the theoretical analysis of phase retrieval (PR) and it is adopted in classic prior works such as [54,12], and in the papers [56,89] that study MPR under sparse priors, as well as in the papers [30,36,37,45] that study PR under generative priors. We agree that non-Gaussian measurement models such as sub-sampled Fourier measurements are more practical, and the extension to these measurement models is a very interesting future direction (we will mention this in the Conclusion and Future Work section). However, it is beyond the scope of the current work.

(**The initialization condition of Theorem 1**) We follow [47] to assume the initialization condition $\mathbf{x}^T\mathbf{w}^{(t_{0})} \ge c_0$, which basically assumes weak recovery of the signal (and as mentioned by Reviewer c4oX, this does not seem to be an issue in practice and we do not force such a weak correlation to exist in the numerical simulations). As far as we can tell, it appears to be the mildest initialization condition that we can assume for practical spectral initialization for PR with generative priors. The reason is as follows: Practical spectral initialization methods for sparse PR/MPR typically first estimate the support of the signal and then perform the power method on the submatrix corresponding to the estimated support (see, e.g., [54,9,38,83,89]), or relax the problem to a (convex) semidefinite program (see [56]). Unfortunately, for a generative model, both ideas no longer work since we cannot estimate a set that plays a similar role as the support, and without further assumptions, the problem cannot be relaxed to a convex optimization problem.

(**Comparison with the Bayes-optimal rate/error derived in [3]**) Our results seem to be not directly comparable to those in [3] due to significant differences in the settings. More specifically, in [3]: (i) An AMP algorithm is proposed, and a neural network with i.i.d. Gaussian weights and no offsets is assumed (whereas we only impose the Lipschitz continuity assumption on the generative model. The activation function of our pre-trained neural networks is not restricted to be ReLU and there are offsets). (ii) Asymptotic analysis (not fully rigorous) is given under the high-dimensional regime with $m/n$ being fixed (whereas we provide a rigorous analysis with no restrictions on $m/n$). (iii) The noiseless PR model is focused on (whereas we study the MPR model).

(**Technical novelties**) Our analysis builds on works such as [56,47,70], but we believe that these techniques are combined and extended in a novel manner, with distinct proofs. For instance:

(a) In [56], the estimator is constructed via refining the solution of a semidefinite program by $\ell_1$-regularized regression. In contrast, we use the projected power method for spectral initialization, and then an iterative procedure is performed to refine the initial guess. We believe that for generative models, an iterative procedure is much more practical since the corresponding optimization problem is non-convex and cannot be solved exactly, and we believe that the direct study of generative priors adds significant value to existing approaches based on convex relaxations.

(b) We make use of the method proposed in [47] in our Step 1, but we believe that Theorem 1 is an original and valuable contribution relative to [47] (as mentioned by Reviewer VWWv). In particular, in the proof of Theorem 1, we need to carefully deal with the effect of statistical model misspecification. This requires proving Lemmas 4, 5, and 6, along with a more powerful concentration inequality for sub-Weibull random variables of order $\frac{1}{2}$ (in comparison, in [47], sub-exponential concentration is sufficient).

(c) A projected descent algorithm has been proposed in [70] for linear measurements with generative priors. We remark that one major difference between our Step 2 and the algorithm in [70] is that we need to take the scale factor into account (for the algorithm in [70], there is no scale factor), and we observed from numerical experiments that the scale factor plays an important role. For example, if it is not varying with $t$ (e.g., fixing it as $\hat{\nu}^{(0)}$, instead of $\hat{\nu}^{(t)}$), the reconstruction performance will be significantly worse (we will present corresponding numerical results in the revised version). In addition, the authors of [70] only provide a simple analysis for noiseless linear measurements, whereas we provide a much more complicated analysis for the general MPR model.

---

### Meta-Review · Area_Chair_HD3D · 2022-08-26

**Recommendation:** Accept
**Confidence:** Certain

**Metareview:**

In this paper, the authors study the standard phase retrieval problem, in the case where the signal is assumed to come from a generative model prior. In particular, they propose an algorithm that starts with a spectral method followed by an iterative approach. The authors provide two theorems giving guarantees on the performance of each step of the algorithm and illustrate how their procedure performs with respect to some previous algorithms.

All reviewers were found to judge positively the work of the authors,  finding the paper clear, and well organized, and discussing honestly both the advantages and the limitations of their methods and theorems. The reviewers also found the answer to their questions during the rebuttal phase satisfactory.

**Award:**

No

---

### Decision · Program_Chairs · 2022-09-14

Accept